# Underestimations in the In Silico-Predicted Toxicities of V-Agents

Georgios Pampalakis

Laboratory of Pharmacology, School of Pharmacy, Aristotle University of Thessaloniki, 54124 Thessaloniki, Greece; gpampalakis@pharm.auth.gr

**Abstract:** V-agents are exceedingly toxic nerve agents. Recently, it was highlighted that V-agents constitute a diverse subclass of compounds with most of them not extensively studied. Although chemical weapons have been banned under the Chemical Weapons Convention (CWC), there is an increased concern for chemical terrorism. Thus, it is important to understand their properties and toxicities, especially since some of these agents are not included in the CWC list. Nonetheless, to achieve this goal, the testing of a huge number of compounds is needed. Alternatively, in silico toxicology offers a great advantage for the rapid assessment of toxic compounds. Here, various in silico tools (TEST, VEGA, pkCSM ProTox-II) were used to estimate the acute oral toxicity (LD50) of different V-agents and compare them with experimental values. These programs underestimated the toxicity of V-agents, and certain V-agents were estimated to be relatively non-toxic. TEST was also used to estimate the physical properties and found to provide good approximations for densities, surface tensions and vapor pressures but not for viscosities. Thus, attention should be paid when interpreting and estimating the toxicities of V-agents in silico, and it is necessary to conduct future detailed experiments to understand their properties and develop effective countermeasures.

**Keywords:** in silico toxicology; V-agents; Toxicity Estimation Software Tool (TEST); ProTox-II





## 1. Introduction

V-agents are exceedingly toxic organophosphate nerve agents. They are oily liquids with low vapor pressure. VX and RVX are the most widely known and studied V-agents. Nonetheless, a recent thorough literature surveillance showed that V-agents constitute a large family of nerve agents with seven different subclasses (Figure 1). Most of these agents (including variants of VX with different constituents) are relatively non-studied, and some of these agents (e.g., EA-1576 and VP) are not included in the Chemical Weapons Convention [1]. Sarin, a G-type nerve agent, was used in the terrorist attacks organized by the religious cult Aum Shinrikyo in Japan in 1994 and 1995 [2]. Afterwards, the public concern on the threat posed by nerve agents has raised significantly especially in the recent years after sarin was used in 2013 and 2017 in the Syrian civil war [3,4], the assassination of Kim Jong Nam with a binary form of VX [5], the series of assassinations in UK [6] and the assassination attempt of Alexei Navalny in Russia, with Novichok agents [7].

In this direction, the other V-agents that have not been extensively studied pose great threat with unknown consequences and potential medical countermeasures. Therefore, it is important to understand their toxicity and properties that relate to dissemination [8]. To accomplish this endeavor, the testing of a huge number of compounds is required. In silico toxicology approaches facilitate the determination of acute toxicity (LD50) of multiple compounds in a short time and without using animals. Thus, in silico toxicology is an important field for the assessment of potential chemical warfare agents. Previously, the TEST software was used to predict the acute oral toxicity in rats of Novichok nerve agents [9]. The ProTox-II program was developed as an alternative to TEST and is an online platform for the prediction of acute oral rat toxicity [10]. In silico methods have

also been applied to predict the hydrolysis and biotransformation of Novichok agents [11]. In this direction, other theoretical studies used thermodynamics and Density Functional Theory (DFT) to predict which Novichok agent would be more reactive towards the enzyme acetylcholinesterase (AChE) [12]. Here, we have applied TEST and ProTox-II to predict the potential acute oral toxicities of several types of V-agents that belong to all the described subclasses. Notably, many V-agents were predicted to be relatively non-toxic, especially those agents that contained more complex and bulky substituents. Thus, it is suggested that TEST and ProTox-II cannot provide reliable estimates on the potential toxicity of V-agent, and the results should be interpreted with extreme caution. Further, VEGA and pkCSM were applied for acute oral LD50 prediction in rats that again provided underestimated values, especially the program pkCSM. In addition, TEST was applied to predict the physical properties of V-agents and specifically their density, surface tenson, viscosity and vapor pressure. In this case, many of the properties were in good approximation with the experimental data except for viscosities.

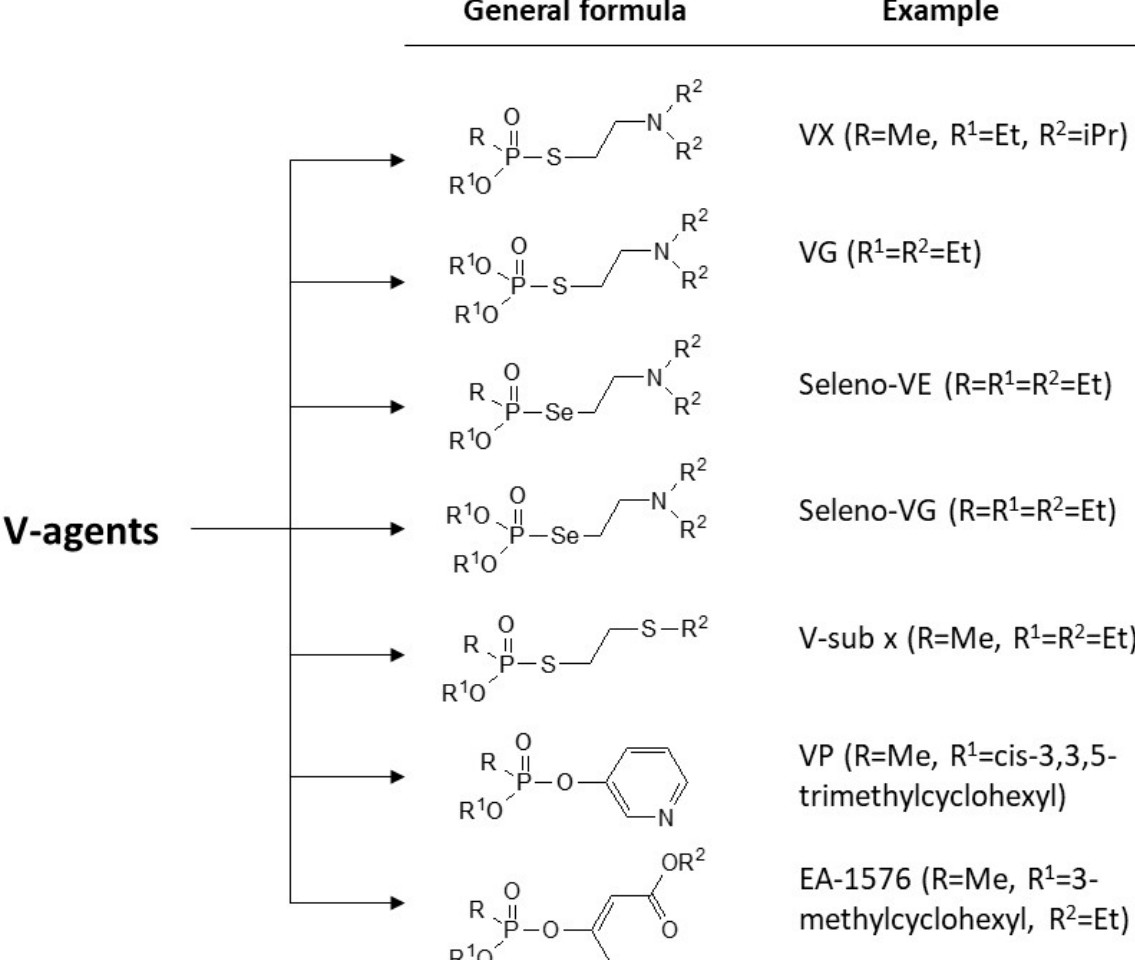

**Figure 1.** Classification of V-agents. The classification was performed according to [1]. The general formula of each subclass and a representative example is given.

## 2. Methods

### 2.1. TEST

Toxicity Estimation Software Tool (TEST) as stated in its manual is a "Java application to estimate toxicities and physical properties from molecular structure". It is a publicly available QSAR software developed for the US Environmental Protection Agency

(EPA). Here, version 5.1.2 was used (https://www.epa.gov/chemical-research/toxicity-estimation-software-tool-test, downloaded on 10 June 2023). The chemicals can be entered either through their SMILES code, CAS number, name or through the draw formula window. The program allows the prediction of the acute rat oral LD50 for the query compound with three different approaches: consensus, hierarchical clustering, nearest neighbor. Further details on TEST were recently published [9]. Also, it allows the prediction of various physical properties including density, surface tension, viscosity and vapor pressure using the following approaches: consensus, hierarchical clustering, nearest neighbor, group contribution and single model (additional model for viscosity). For comparison, the consensus result was used.

### 2.2. ProTox-II

ProTox-II is a web-based tool for the prediction of acute rat oral toxicity (https://tox-new.charite.de/protox_II/index.php?site=compound_search_similarity, accessed on 1 July 2023). The toxicity models are based on two-dimensional analysis of similarities between the query compounds and compounds present in the library with known LD50s [10,13]. The chemicals can be entered by name, SMILES or drawn online. It is reported that ProTox-II performs comparatively better than TEST [10].

### 2.3. VEGA

The VEGA QSAR (quantitative-structure activity relationship) is a free to download application (https://www.vegahub.eu/download/vega-qsar-download, downloaded on 11 October 2023) for the prediction of various biological parameters of a given chemical that includes the acute rat oral LD50s [14]. Other parameters that may be predicted include skin and eye irritation, carcinogenicity, mutagenicity (Ames test), etc. The chemicals are introduced with their SMILES code.

### 2.4. pkCSM

The pkCSM is a web-based tool for the prediction of absorption–distribution–metabolism–excretion–toxicity (ADMET) properties of chemicals (https://biosig.lab.uq.edu.au/pkcsm/prediction, accessed on 11 October 2023). The output includes predictions of acute rat oral LD50s [15]. The chemicals are introduced with their SMILES code. The predicted toxicities are reported as $mol \cdot kg^{-1}$ and were converted here to $mg \cdot kg^{-1}$ using their molecular weight.

### 2.5. Experimental Toxicities and Physical Properties of V-Agents and Pesticides

All experimental toxicities reported here are acute oral LD50s in rats unless otherwise stated, e.g., for VP (percutaneous toxicity in rabbits was available). The toxicities of pesticides parathion, demeton-S and E-mevinphos, as well as the toxicity of VG, were retrieved from Pubchem (https://pubchem.ncbi.nlm.nih.gov/; last accessed on 15 July 2023). The toxicities of VR and EA-2192 were obtained from [16,17], respectively. All other toxicities of V-agents were reviewed and reported in a recent article [1]. LD50s are given in $mg \cdot kg^{-1}$. The physical properties of V-agents were also obtained from a recent review article [1]. The physical properties of demeton-S were obtained from Pubchem. Since the prior data on viscosities were in cS, here, all data are given in cP (the result of TEST) using the formula cP = cS·density.

## 3. Results and Discussion

Initially, a series of V-agents belonging to all the described subclasses, were analyzed for prediction of oral LD50 using TEST as shown in Table 1. For some of these compounds, the oral LD50 in rats has been reported; therefore, these compounds provide a means for direct comparison. Specifically, VX exhibits oral LD50 in rats of 0.085 $mg \cdot kg^{-1}$, while TEST predicts a toxicity of 1.95 $mg \cdot kg^{-1}$, which is 23 times higher. In accordance, the toxicities of the subclass 1 agents VM, EA-1728 (EA: Edgewood Arsenal), EA-1694 and EA-1699 exhibit an underestimation ratio (predicted LD50/experimental LD50) of 5.1, 105.7,

12.2 and 7.9 times, respectively. The most striking underestimation occurred for EA-1728 that contains an isopropoxy group in the place of ethoxy group for VX. To this end, it should be noted that EA-3148, which is more potent than VX and shows signs of intoxication in human volunteers even when administered iv at 0.000115 mg·kg$^{-1}$ [18], had a predicted LD50 5.92 mg·kg$^{-1}$, which is even lower that the pesticide mevinphos (Table 1). Notably, for additional comparison, parathion has an oral LD50 in rats 2 mg·kg$^{-1}$, which is again higher than the predicted by TEST for the EA-3148. Therefore, the predicted toxicity of EA-3148 does not classify it as a potential chemical warfare agent, which is not the actual case. Overall, it appears that increasing the number of carbon atoms in the alkoxy moiety negatively impacts the estimation of oral LD50 by TEST. To this end, it should be noted that for VR, there are two completely different reported experimental oral rat LD50 values of 0.020 and 1.402 mg·kg$^{-1}$ [16,19], and thus, no attempt was performed to correlate the experimental values with the in silico predicted values.

When the structure of the organophosphate becomes more complex as demonstrated with VP, the prediction of toxicity is severely affected. First, the program does not allow to predict any differences in the oral toxicity of the VP derived from cis-3,3,5-trimethylcyclohexanol compared to VP derived from the trans alcohol, although it is known that the cis analog is the more toxic and "weapons grade" material [20]. More importantly, the estimated oral LD50 of VP is 1837.54 mg·kg$^{-1}$, which defines it as slightly toxic and thus not as a potential chemical warfare agent.

In Table 1, the predicted oral LD50s of the pesticides demeton-S and mevinphos (mentioned above) have also been included since these compounds provide the chemical scaffold for designing the nerve agents V-sub x and EA-1576 by converting the phosphorylated pesticides to the phosphonylated analogues. As shown in Table 1, the predicted oral toxicity of V-sub x is approximately the same as demeton-S. Accordingly, the prediction of oral toxicity for the compound EA-1576 classifies it as moderately toxic and with significantly lower toxicity (>30 times) compared to E-mevinphos. Although, no LD50 data have been published for EA-1576, it is known that it is a chemical warfare agent and a phosphonylated analog of E-mevinphos, a substitution that results in significantly higher toxicity. Further, TEST did not predict difference between the E- and the Z-isomers of EA-1576, while it is known that the E-isomer is more toxic [21]. To this end, E-mevinphos was predicted to be slightly less toxic than the Z-mevinphos, but experiments have shown that the reverse is true and when administered ip in rats, the E-isomer exhibits 100-fold higher potency than the Z [22]. Finally, although exact LD50 values have not been published, the cyclohexyl analogue of VP has been reported to be less toxic than the VP [20]. Notable, TEST predicts the opposite.

For all the above comments, the TEST prediction made by consensus has been used. Although there are some variations when the prediction is based on hierarchical clustering or on nearest neighbor, the basic conclusion remains the same and TEST underestimates the acute oral toxicity of several V-type nerve agents. It was also tried to predict the toxicities of the phospho(n/r)ylated selenothiocholines (e.g., seleno-VE), but at the moment, TEST cannot provide data for selenium containing compounds.

**Table 1.** Predicted and experimental rat oral LD50s for V-agents.

| Agent | Formula | Predicted LD50, mg·kg⁻¹ | | | | | | Exp [a] LD50, mg·kg⁻¹ | Other LD50, mg·kg⁻¹ |
|---|---|---|---|---|---|---|---|---|---|
| | | HC [a] (TEST) | NN [a] (TEST) | Con [a] (TEST) | ProTox | VEGA | pkCSM | | |
| VX | $MeP(O)(OEt)SCH_2CH_2NiPr_2$ | 1.44 | 2.67 | 1.95 | 1 | 3.78 | 778,000 | 0.085 0.122 | |
| VS | $EtP(O)(OEt)SCH_2CH_2NiPr_2$ | 5.14 | 9.86 | 7.12 | 1 | 3.87 | 803,400 | | |
| VE | $EtP(O)(OEt)SCH_2CH_2NEt_2$ | 4.05 | 1.43 | 2.41 | 1 | 3.31 | 795,000 | | |
| VM | $MeP(O)(OEt)SCH_2CH_2NEt_2$ | 0.86 | 1.35 | 1.08 | 1 | 1.49 | 747,000 | 0.212 | |
| VR | $MeP(O)(OiBu)SCH_2CH_2NEt_2$ | 0.72 | 1.51 | 1.05 | 1 | | 839,000 | 0.020 1.402 | |
| EA-1728 | $MeP(O)(OiPr)SCH_2CH_2NiPr_2$ | 1.48 | 23.59 | 5.92 | 1 | 3.9 | 788,800 | 0.056 | |
| EA-1763 | $MeP(O)(OPr)SCH_2CH_2NiPr_2$ | 12.12 | 9.86 | 10.93 | 1 | 3.91 | 817,000 | | |
| EA-1694 | $EtP(O)(OEt)SCH_2CH_2NMet_2$ | 1.71 | 1.27 | 1.48 | 1 | 1.43 | 697,700 | 0.121 | |
| EA-1699 | $MeP(O)(OEt)SCH_2CH_2NMet_2$ | 0.31 | 2.95 | 0.96 | 1 | 1.32 | 648,800 | 0.122 | |
| EA-3148 | $MeP(O)(Ocp)SCH_2CH_2NEt_2$ [b] | 1.90 | 18.43 | 5.92 | 1 | 79.07 | 885,000 | | |
| CVX | $MeP(O)(OBu)SCH_2CH_2NEt_2$ | 0.99 | 1.51 | 1.23 | 1 | 3.74 | 849,700 | | |
| VG | $(EtO)_2P(O)SCH_2CH_2NEt_2$ | 14.01 | 3.70 | 7.20 | 3 | 3.31 | 888,000 | 3.3 | |
| VP (cis) | See Figure 1 | 991.24 | 3406.41 | 1837.54 | 9333 | 782.51 | 821,000 | | 0.0818 (rabbit pc) |
| VP (trans) | | 991.24 | 3406.41 | 1837.54 | 9333 | | | | |
| V sub x | $MeP(O)(OEt)SCH_2CH_2SEt$ | 7 | 5.28 | 6.08 | 3 | 7.06 | 690,000 | | |
| EA-1576 (E) | See Figure 1 | 166.21 | 49.59 | 90.79 | 44 | | 885,000 | | More toxic than Z |
| EA-1576 (Z) | | 166.21 | 49.59 | 90.79 | 44 | | | | |
| EA-2192 | $MeP(O)(OH)SCH_2CH_2NiPr_2$ | 2.41 | 11.68 | 5.30 | 826 | 3.43 | 565,000 | 0.63 | |
| Cyclohexyl-VP | VP with cyclohexyl instead of 3,3,5-trimethylcyclohexyl group | 207.40 | 2924.27 | 778.77 | | 720.84 | 726,000 | | Less toxic than cis-VP |
| Demeton-S | $(EtO)_2P(O)SCH_2CH_2SEt$ | 8.43 | 4.7 | 6.29 | | 1.49 | 820,000 | 1.5 | |
| E-mevinphos | $(MeO)_2P(O)OC(CH_3) = CHCOOMe$ | 4.46 | 14.16 | 7.95 | | 3.02 | 645,000 | 3 | |
| Z-mevinphos | | 4.46 | 6.67 | 5.45 | | 3.02 | | | |

[a] HC: hierarchical clustering; NN: nearest neighbor; Con: consensus; Exp: experimental.; [b] Ocp: o-cyclopentyl.

When ProTox-II was used to predict the oral LD50, it was found that all the phosphonylated thiocholines (VX, VS, VE, VM, VR, EA-1728, EA-1694, EA-1699, EA-3148 and CVX) had the same LD50 equal to 1 mg·kg$^{-1}$. Nonetheless, a very good approximation of the experimental LD50 for VG was given by ProTox-II. As with the case of TEST, when bulky organic moieties are introduced in the V-agent molecule, the program failed to predict the high toxicity of the agent. Specifically, it was found that both cis and trans VP displayed a predicted LD50 of 9333 mg·kg$^{-1}$, values that are identical to the cyclohexyl analog of VP and most importantly that classify them almost as being non-toxic compounds. Further, ProTox-II failed to show any difference between the toxicity of E and Z-isomers of EA-1576 as well as between the E and Z-mevinphos. Additionally, it predicted that mevinphos is four times more toxic than EA-1576. In the same direction, ProTox-II predicted an LD50 of 2 mg·kg$^{-1}$ for the pesticide demeton-S but it predicted lower toxicity of the phosphonylated analog V-sub x (3 mg·kg$^{-1}$). Also, for the hydrolysis product of VX, EA-2192, which is known to be an exceedingly toxic compound with an acute oral rat LD50 of 0.63 mg·kg$^{-1}$ [17], ProTox-II predicts a toxicity of 826 mg·kg$^{-1}$, which is 1300 times higher.

Using ProTox-II, seleno-VE and seleno-VG are estimated to have an LD50 of 3 mg·kg$^{-1}$. Notably, the phosphor(n/r)ylated selenocholines are more toxic than the respective thiocholines, and although no oral LD50 data are available for seleno-VE and seleno-VG, the estimation of 3 mg·kg$^{-1}$ may be significantly underestimated from the experimental values for the following reasons: (a) the experimental sc LD50 values of seleno-VE and seleno-VG in mice are 0.021 and 0.060 mg kg$^{-1}$ [23], (b) there is a failure to predict differences between the LD50 of seleno-VE and seleno-VG by ProTox-II, when it is known that phosphonates exhibit higher toxicity than their phosphorylated analogues, and (c) the predicted LD50 of seleno-VE is higher than VE, while seleno-VG is predicted to be equally toxic to VG.

In addition, the programs VEGA and pkCSM were used to predict the acute rat oral LD50 of V-agents. Regarding the pkCSM, a well-established ADMET program [24], all prediction showed LD50s > 500,000 mg·kg$^{-1}$, which points to non-toxic compounds. On the other hand, VEGA provides results that are somehow analogous to TEST and ProTox-II. For example, increasing the complexity of alkoxy moiety as illustrated with EA-3148, which has a cyclopentyl group, results in severely underestimated oral LD50 values (Table 1). Further, demeton-S is predicted to be more toxic that the V-sub x, and cyclohexyl-VP is predicted to be slightly more toxic that VP, which as analyzed previously are known not to be correct.

Taking all data together, it is obvious that there is no way to provide a prediction of the LD50s based on a consensus value generated by these programs, since all estimations highly deviate from experimental values.

Then, TEST was used to estimate the physical properties of V-agents and specifically the density, the surface tension, the viscosity and the vapor pressure (Table 2). For most of the cases, viscosities and surface tensions could only be predicted with the nearest neighbor method and the estimation of viscosities showed large variations compared to experimental values. Regarding density, EA-1576 is predicted to have the highest density 1.16 g·ml$^{-1}$ and it is indeed the compound with the highest experimentally reported density of 1.0829 g·ml$^{-1}$. Furthermore, it is known that conversion of a phosphorylated thiocholine to phosphonylated thiocholine reduces the density as demonstrated for VG and VE (predicted 1.09 and 1.08 g·ml$^{-1}$, respectively, with values experimental 1.04577 and 1.0180 g·ml$^{-1}$, respectively). Regarding vapor pressure, V-agents are predicted to be low-volatility liquids as expected (vapor pressure < 10·10$^{-4}$ mmHg).

**Table 2.** Predicted and experimental physical properties of V-agents.

| | Density @25 °C | | | | | Surface Tension @25 °C (dyn/cm) | | | | | Viscosity @25 °C (cP) | | | | | | Vapor Pressure @25 °C ($\times 10^{-4}$) | | | | |
|---|---|---|---|---|---|---|---|---|---|---|---|---|---|---|---|---|---|---|---|---|---|
| | HC | G | NN | Con | Exp | HC | G | NN | Con | Exp | HC | G | NN | SM | Con | Exp | HC | G | N | Con | Exp |
| VX | 1.03 | 0.94 | 1.20 | 1.06 | 1.0083 | | | 27.80 | 31.6 | | | | 5.21 | | | 10.04 | 6.59 | 3,71 | 0.214 | 1.75 | 8.78 |
| VS | 1.02 | 0.92 | 1.24 | 1.06 | 1.0016 | | | 27.85 | 29.9 | | | | 5.21 | | | 9.37 | 0.937 | 1.33 | 2.94 | 1.54 | |
| VE | 1.03 | 1.01 | 1.20 | 1.08 | 1.0180 | | | 27.80 | 29.5 | | | | 5.21 | | | 5.54 | 3.14 | 2.16 | 2.94 | 2.71 | |
| VM | 1.04 | 1.02 | 1.20 | 1.09 | 1.0311 | | | 27.64 | 31.2 | | | | 5.21 | | | 6.03 | 5.54 | 6.10 | 2.94 | 4.63 | |
| VR | 1.02 | 0.96 | 1.20 | 1.06 | 1.0065 | | | 28.06 | 26.9 | | | | 5.21 | | | 8.39 | 5.44 | 1.7 | 2.94 | 3 | 6.3 |
| EA-1728 | 1.02 | 0.90 | 1.24 | 1.05 | 0.9899 | | | 27.28 | 29.2 | | | | 5.21 | | | 11.28 | 6.36 | 2.96 | 2.94 | 3.81 | |
| EA-1763 | 1.02 | 0.92 | 1.24 | 1.06 | 0.9973 | | | 28.06 | 30.2 | | | | 5.21 | | | 11.26 | 2.74 | 1.33 | 2.94 | 2.21 | |
| EA-1694 | 1.07 | 1.04 | 1.13 | 1.08 | 1.0453 | | | 27.58 | 31.5 | | | | 5.21 | | | 5.14 | 6.61 | 17.3 | 2.94 | 6.95 | |
| EA-1699 | 1.07 | 1.04 | 1.13 | 1.08 | 1.0600 | | | 27.58 | 32.0 | | | | 5.21 | | | 5.62 | 7.37 | 48.8 | 2.94 | 10.2 | |
| EA-3148 | 1.07 | 1.06 | 1.23 | 1.12 | 1.05 | | | 28.20 | | | | | 7.69 | | | 2.06 | 0.375 | 0.175 | 2.94 | 0.578 | 4 |
| CVX | 1.02 | 0.99 | 1.20 | 1.07 | 1.0125 | | | 28.06 | 22.68 | | | | 5.21 | | | 9.41 | 4.31 | 7.63 | 2.94 | 2.13 | 2.5 |
| VG | 1.06 | 1.01 | 1.20 | 1.09 | 1.04557 | | | 27.80 | 31.0 | | | | 5.21 | | | 4.96 | 3.15 | 1.69 | 2.94 | 2.51 | |
| VP (cis) | 1.10 | 1.09 | 1.13 | 1.10 | 1.023 | | | 28.06 | 30.4 | 31.98 | | | 16.92 | 31.98 | 25.86 | 30.28 | 0.296 | 0.548 | 0.00266 | 0.0755 | |
| VP (trans) | 1.10 | 1.09 | 1.13 | 1.10 | | | | 28.06 | | 31.98 | | | 16.92 | 31.98 | | | 0.296 | 0.548 | 0.00266 | 0.0755 | |
| V sub x | 1.13 | 1.15 | 1.17 | 1.15 | | | | 27.64 | | | | | 2.58 | | | | 0.773 | 6.86 | 3.83 | 2.73 | |
| EA-1576 (E) | 1.11 | 1.13 | 1.24 | 1.16 | 1.0829 | 30.98 | 37.24 | 28.07 | 32.08 | 32.4 | | | 5.21 | | | 25.23 | 0.0294 | 0.0831 | 0.331 | 0.0931 | |
| EA-1576 (Z) | 1.11 | 1.13 | 1.24 | 1.16 | | 30.98 | 37.24 | 28.07 | 32.08 | | | | 5.21 | | | | 0.0294 | 0.0831 | 0.331 | 0.0931 | |
| EA-2192 | 1.11 | 1.07 | 1.20 | 1.12 | | | | 27.64 | | | | | 5.21 | | | | | | | 0.395 | |
| Cyclohex-VP | 1.15 | 1.21 | 1.13 | 1.17 | | | | 28.07 | 32.45 | | | | 16.92 | 32.45 | 26.12 | | 0.172 | 0.124 | 0.00266 | 0.0827 | |
| Demeton-S | 1.15 | 1.14 | 1.20 | 1.16 | 1.132 @21 °C | | | 27.20 | | | | | 5.21 | | | | 1.19 | 1.90 | 3.94 | 2.07 | 2.6 @20 °C |
| E-mevinphos | 1.22 | 1.17 | 1.23 | 1.21 | | 27.19 | 34.56 | 28.64 | 33.46 | | | | 5.21 | | | | 65.3 | | 1.44 | 9.69 | |
| Z-mevinphos | 1.22 | 1.17 | 1.22 | 1.21 | | 37.12 | 34.56 | 28.64 | 33.46 | | | | 5.21 | | | | 65.3 | | 1.44 | 9.69 | |

This indicates that the tools to predict the physicochemical properties are obviously different from the ones that are used for the prediction of oral LD50, and at the same time, they are much more accurate and valid. A reason for the more accurate results of the physicochemical properties may be the fact that there are several mathematical equations that relate the physical properties of a chemical with molecular characteristics [25].

To this end, it should be noted that the in silico methodologies offer a great advantage in the assessment of acute oral LD50 of new acetylcholinesterase reactivators that are designed for administration against nerve agent poisoning [26]. These reactivators are oximes and there are a large variety of oxime reactivators that must be assessed for their safety. The need for new reactivators comes from the fact that the phospho(n/r)ylated AChE may be subjected to rapid aging or when reacting with nerve agents like VR that contain bulky alkoxy groups, these groups stereochemically prohibit the reactivator from reaching the active site [27]. Further, the design of new oximes should take into consideration the need to penetrate the blood–brain barrier that increases the chemical variability and thus necessitates the pre-screening/pre-selection of the compounds for potential toxicity [28]. On the other hand, molecular docking and molecular dynamics have been used to assess the efficacy or reactivators against inhibited AChE [29].

QSAR tools are constantly becoming important players in toxicology and have been extensively applied in many fields even for the prediction of nanoparticle toxicity (nanotoxicity) [30]. QSAR are mathematical models that correlated structure with biological activity. It is therefore evident that the closest the structure of the query compound is to the structure of a compound in a database, the more accurate the prediction will be. New improvements in QSAR include the incorporation of Density Functional Theory (DFT) elements and molecular docking [31]. Recently, machine learning methods have also been applied in toxicology in silico [32]. Machine learning has been applied for the prediction of Novichok oral LD50 [33] and could provide the basis for a future tool to assess the acute toxicity of V-agents. Interestingly, machine learning has also been applied for the determination of the vapor pressure of Novichok agents [33].

In silico studies have also been used to predict the IR and Raman spectra of Novichok agents [34]. Further, semi-empirical methods and molecular modeling have been used to provide an explanation for the putative very high toxicity of Novichok agents that is supposed to be higher the VX [35]. These data are in contrast with the ones obtained using TEST [9]. Future experimental work on Novichok will delineate which theoretical study corresponds better to the experimental values. This contradiction also points to the need for more experimental work on Novichoks.

The need for in silico toxicology comes from the fact that a slight modification (new substituent) may significantly alter toxicity, and countermeasures. To test all the putative compounds is a very difficult and time-consuming task without taking into consideration the ethical issues derived from the huge number of experimental animals that are needed.

A potential limitation of the present study is based on the fact that for many of the V-agents reported here, their oral acute LD50s in rats have not been experimentally determined. This does not facilitate the calculation of a potential deviation factor between the estimated and the experimental values. The strength of the study is the fact that it highlights the need for further experimentation on the toxicological properties of V-agents with the aim to design effective countermeasures and improved QSAR programs.

Through this study, the limitations on predicting oral rat LD50s of these programs was revealed. Specifically, the LD50s of V-agents cannot reliably estimated. However, TEST can be applied for the determination of physicochemical properties and pkCSM for ADME properties. More detailed studies should also be carried out in the future using compounds that have known LD50s as queries to determine which classes of chemicals could be reliably estimated.

## 4. Conclusions

V-agents constitute a diverse class of nerve agents that pose a major threat in potential chemical terrorism events or asymmetric attacks. Here, in silico toxicology application platforms (TEST, developed for US EPA, and ProTox-II, which is reported to perform comparatively better than TEST, VEGA and pkCSM) were used to assess their ability to estimate the oral rat LD50 of V-agents. All these tools were found to underestimate the toxicity of V-agents and in some cases even predict that the agent is almost non-toxic. Thus, LD50 prediction for V-agents should be cautiously used since they may significantly deviate from the actual values and further experimental works should be carried out to understand the chemical and toxicological properties of these agents. These findings call for the development of new, potentially more specialized QSAR tools that are based on libraries with a large variety of nerve agents belonging to different subclasses and encompassing a diversity of substituents. Importantly, tools that will predict percutaneous toxicity should be developed as well, since V-agents constitute mainly a percutaneous hazard as they are oily liquids with low vapor pressure. To successfully accomplish this task, further experimentation is required on the new types of V-agents that belong to different subclasses. Once the experimental LD50s are established for a small number of V-agents belonging to different subclasses, these could be applied for the generation of a new database for QSAR predictions. This will reduce the number of animals required to assess every potential agent (only representative agents are required), and further, it will reduce the time required for their assessment. However, it should be taken into consideration that due to ethical issues, the determination of experimental LD50s may be completely deemed unnecessary. In this case, the use of IC50s for AChE could provide an alternative method for the generation of QSAR equations that will provide more accurate LD50 predictions. To accomplish this task, interlaboratory standardization of AChE inhibition assays is required.

**Funding:** This research received no external funding.

**Institutional Review Board Statement:** Not applicable.

**Informed Consent Statement:** Not applicable.

**Data Availability Statement:** All data analyzed here are included in the manuscript.

**Conflicts of Interest:** The author declares no conflict of interest.

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
