# Peer review of "Underestimations in the In Silico-Predicted Toxicities of V-Agents"

_jox, doi:10.3390/jox13040039_

Round 1

Reviewer 1 Report

This manuscript deals with the comparison of software-based prediction of LD50 of nerve agents of the V-subclass family with a careful review of the literature.
The prediction software studied here, TEST and Pro-Tox-II, allow also a prediction of the physicochemical properties of the different V-agents.

Most of the predicted rat oral LD50s are over-estimated by these softwares compared to the values reported in the literature, demonstrating the weakness of these predictions, especially for toxics whose privileged intoxication route would be the percutaneous one. For some agents, the predicted values were such that some could be considered almost non-toxic, while highly toxic experimentally. Thus use of such software for the prediction of the toxicity of these agents is clearly not recommended.

On the contrary, the prediction of the physicochemical properties of the different agents is mostly in agreement with data available in the literature, except for viscosity, which may be more difficult to predict.

This manuscript is based on a very simple methodology as it relies on the comparison of published data and prediction software results. It would have been interesting to propose an upgrade or an alternative software for such LD50 predictions. The results are obvious but a careful review of the literature was necessary and is accomplished in this manuscript.

Author Response

Reviewer 1

This manuscript deals with the comparison of software-based prediction of LD50 of nerve agents of the V-subclass family with a careful review of the literature.
The prediction software studied here, TEST and Pro-Tox-II, allow also a prediction of the physicochemical properties of the different V-agents.

Most of the predicted rat oral LD50s are over-estimated by these softwares compared to the values reported in the literature, demonstrating the weakness of these predictions, especially for toxics whose privileged intoxication route would be the percutaneous one. For some agents, the predicted values were such that some could be considered almost non-toxic, while highly toxic experimentally. Thus use of such software for the prediction of the toxicity of these agents is clearly not recommended.

On the contrary, the prediction of the physicochemical properties of the different agents is mostly in agreement with data available in the literature, except for viscosity, which may be more difficult to predict.

Comment

This manuscript is based on a very simple methodology as it relies on the comparison of published data and prediction software results. It would have been interesting to propose an upgrade or an alternative software for such LD50 predictions. The results are obvious but a careful review of the literature was necessary and is accomplished in this manuscript.

Answer

We would like to thank the Reviewer for the nice comments. We have now expanded the Results and Discussion to provide suggestions on the upgrade of the software.

Reviewer 2 Report

Whereas the work seems to be carefully done, some few points need attention before publication.

1) A deeper discussion considering theoretical methods applied to predict the toxitity could be employed, for instance, QSAR and Machine learning studies. In this case, a mention about new methods, such as:

Environment International Volume 177,  108025 (2023)

Current Analytical Chemistry, Volume 19,  436-439(4) (2023)

2) Another important aspect that could enrich the current work is to discuss and include references considering the toxicological profile of acetylcholinesterase reactivators.

CMC Volume30, 4149-4166 (2023)

Author Response

Reviewer 2

Whereas the work seems to be carefully done, some few points need attention before publication.

Comments

1) A deeper discussion considering theoretical methods applied to predict the toxitity could be employed, for instance, QSAR and Machine learning studies. In this case, a mention about new methods, such as:

Environment International Volume 177,  108025 (2023)

Current Analytical Chemistry, Volume 19,  436-439(4) (2023)

Answer

The Results and Discussion sections has been updated and new citations (including the suggested ones) have been incorporated into the manuscript.

2) Another important aspect that could enrich the current work is to discuss and include references considering the toxicological profile of acetylcholinesterase reactivators.

CMC Volume30, 4149-4166 (2023)

Answer

The in silico toxicology to predict the toxicological profile of AChE reactivators has been added into the Results and Discussion section. The suggested paper has been cited along with new papers.
